# From birth to adulthood: An analysis of the Brazilian lancehead (*Bothrops moojeni*) venom at different life stages

**Daniela Miki Hatakeyama**[1,2], **Lídia Jorge Tasima**[1,2], **Nathália da Costa Galizio**[1,2], **Caroline Serino-Silva**[1,2], **Caroline Fabri Bittencourt Rodrigues**[1,2], **Daniel Rodrigues Stuginski**[1], **Sávio Stefanini Sant'Anna**[1], **Kathleen Fernandes Grego**[1], **Alexandre Keiji Tashima**[3], **Erika Sayuri Nishiduka**[3], **Karen de Morais-Zani**[1,2], **Anita Mitico Tanaka-Azevedo**[1,2] *

**1** Laboratório de Herpetologia, Instituto Butantan, São Paulo, Brazil, **2** Interunidades em Biotecnologia, Universidade de São Paulo, Instituto de Pesquisas Tecnológicas, Instituto Butantan, São Paulo, Brazil, **3** Departamento de Bioquímica, Universidade Federal de São Paulo, São Paulo, Brazil

\* amt.azevedo@uol.com.br

**Data Availability Statement:** All relevant data are within the manuscript and its Supporting Information files.

## Abstract

The Brazilian lancehead (*Bothrops moojeni*) has a wide distribution in Brazil and represents a serious public health hazard. Previous works reported that the symptoms of snakebites caused by *B. moojeni* juveniles' bites were mainly related to coagulation, while those caused by adults' bites had a more prominent local damage. In this work, we analyzed the venoms of *B. moojeni* at different life stages to better understand the ontogeny shift in this species. Snakes were grouped by age and sex, and venom pools were formed accordingly. Compositional analyses by one-dimensional electrophoresis (1-DE), chromatography, and mass spectrometry revealed that ontogenetic changes might be mostly related to phospholipase $A_2$ ($PLA_2$) and metalloproteases. Regarding the venoms functional aspect, proteolytic, L-amino acid oxidase, $PLA_2$, and coagulant *in vitro* activities were assayed, but only the first and the last ones showed age-related changes, with the venom of snakes up to 1 year-old displaying lower proteolytic and higher coagulant activities, while those from 2 years-old onward presented the opposite relation. The venoms of 3 years-old snakes were exceptions to the compositional and functional pattern of adults as both venoms presented profiles similar to neonates. Sex-related differences were observed in specific groups and were not age-related. *In vivo* experiments (median lethal dose and hemorrhagic activity) were statistically similar between neonates and adults, however we verified that the adult venom killed mice faster comparing to the neonates. All venoms were mostly recognized by the antibothropic serum and displayed similar profiles to 1-DE in western blotting. In conclusion, the Brazilian lancehead venom showed ontogenetic shift in its composition and activities. Furthermore, this change occurred in snakes from 1 to 2 years-old, and interestingly the venom pools from 3 years-old snakes had particular characteristics, which highlights the importance of comprehensive studies to better understand venom variability.

**Funding:** This work was supported by grants from Coordenação de Aperfeiçoamento de Pessoal de Nível Superior (CAPES) (1736737 to DMH), Conselho Nacional de Desenvolvimento Científico e Tecnológico (CNPq), and (Fundação de Amparo à Pesquisa do Estado de São Paulo (FAPESP) (2018/25899-0 to LJT, 2018/14724-4 to NCG, 2017/20106-9 to AKT, 2017/16908-2 to KdM-Z, and 2018/25786-0 to AMT-A). The funders had no role in study design, data collection and analysis, decision to publish, or preparation of the manuscript.

**Competing interests:** The authors have declared that no competing interests exist.

## Introduction

Classified as a neglected tropical disease by the World Health Organization (in Category A), snakebites are an important public health hazard, especially in tropical and subtropical areas [1]. In 2019, 24,453 envenomings were reported in Brazil [2], with the genus *Bothrops* being responsible for 85.5% of the cases reported [2].

Snake venom is a complex mixture of proteins, peptides, and non-proteic components that has a primary foraging function but also acts as a defense mechanism [3, 4]; additionally, it may help the digestion of preys' tissues [5, 6]. In the case of the *Bothrops* venom, the main effects reported are involved in coagulation disorders, hemorrhage and local tissue damage [7, 8], due to single toxins and/or the synergistic action of the venom's compounds [9]. Most of bothropic venoms are mainly composed by metalloproteases (SVMP), serine proteases (SVSP), phospholipases $A_2$ (PLA$_2$), L-amino acid oxidases (LAAO), and C-type lectins (CTL) [1, 10, 11]. SVMPs are generally responsible for pro- and anti-coagulant disorders and hemorrhage; other effects include edema, pain, blistering, and nephrotoxicity [1, 12]. SVSPs affect the homeostasis acting on the different factors involved in the coagulaltion cascade [13, 14]. Many effects are associated to PLA$_2$s, such as myotoxicity, neurotoxicity, edema, platelet aggregation disturbance, tissue damage, among others [15]. LAAOs are known to be quite labile when exposed to temperature, pH, and even to the proteases present in the venom [16–18] and their main effects are related to cytotoxicity by inducing the release of $H_2O_2$, induction of hemorrhage and apoptosis, and inhibition and induction of platelet aggregation [19]. Lastly, CTLs disrupts blood homeostasis by inducing/inhibiting platelet aggregation or activating/consuming coagulation factors [20].

The Brazilian lancehead (*Bothrops moojeni*, Hoge 1966; Fig 1) inhabits mainly the riparian forest of the Brazilian Cerrado biome (States of Bahia, Distrito Federal, Goiás, Mato Grosso, Mato Grosso do Sul, Minas Gerais, Paraná, Piauí, São Paulo, and Tocantins) and represents a public health problem [21–23]. In general, during the early stages of life, *B. moojeni* present a whitish to yellowish tail tip (Fig 1A) and also a caudal luring behavior. Juvenile *B. moojeni* feeds mainly on small ectotherms (primarily on anurans), but during their development these snakes present an ontogenetic shift on their diets, including small mammals (especially rodents) as an important diet resource [24–26]. The tail tip of the snakes also changes, becoming darker during adulthood (Fig 1B) [25, 26].

Snake venom variability is well known and may be influenced by many factors, such as age, sex, geographic location, season, and captivity conditions; besides, individuals may also differ from each other even if submitted to the same conditions [27–45]. Kouyoumdjian and Polizelli [46] observed that accidents with juvenile *B. moojeni* caused a higher incidence of coagulation disorders than accidents with adult snakes. On the other hand, the latter inflicted more severe local tissue damage in comparison to younger snakes. Later, Furtado and colleagues [28] reported outstanding differences between adult females and their respective offspring in nine species of *Bothrops*, including *B. moojeni*, although individual variation was not considered. The mother's venom of five out of nine species were more lethal than their respective offspring, most newborn venoms presented an electrophoretic profile with bands of higher molecular weight than their respective mothers, caseinolytic activity of all mother's venoms were higher than their newborns, and newborn venoms were more coagulant over plasma and showed higher factor X and prothrombin activation [28].

Besides *B. moojeni*, other species within the genus *Bothrops* also present ontogenetic shift in venom composition and function. For example, the venom of adult specimens of *B. jararaca*, one of the most studied lancehead pit viper, showed predominantly proteolytic activity and local tissue damage when compared to the venom of juveniles, which caused mainly coagulant disorders [47–49]. Similarly, *B. insularis*, *B. atrox*, and *B. asper* are other examples of *Bothrops*

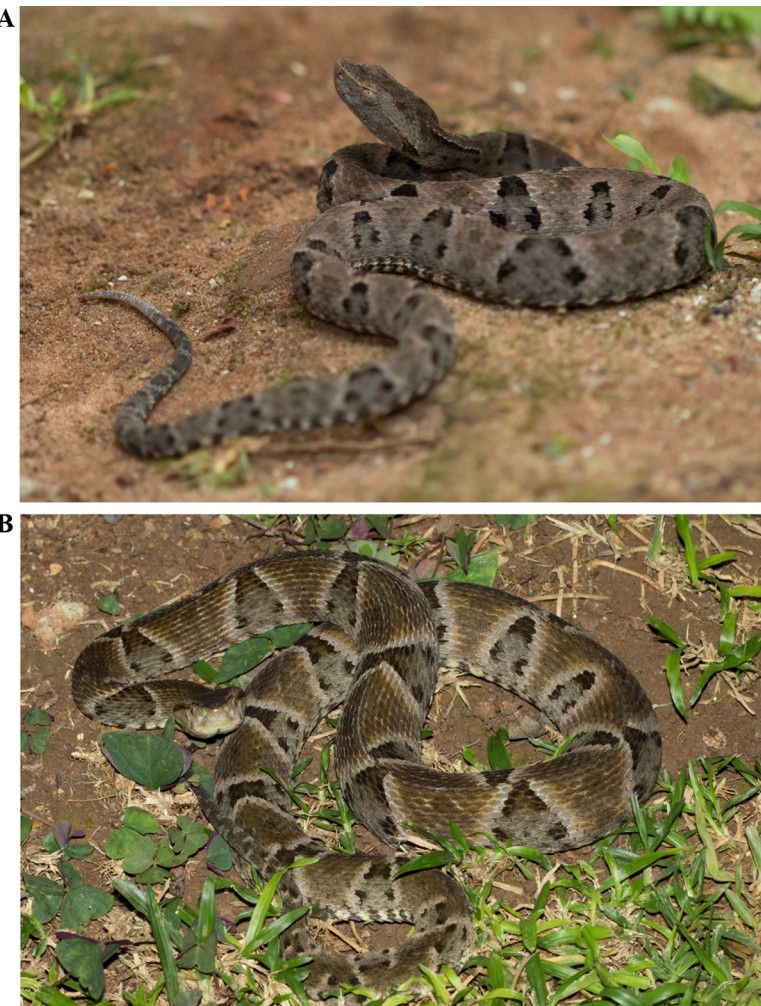

**Fig 1. Juvenile and adult *Bothrops moojeni*.** Specimens of *B. moojeni* snake at juvenile (A) and adult (B) stages. Photos by Sávio Stefanini Sant'Anna.

species which venoms undergoes ontogenetic shift, although *B. insularis* has different feeding habits [41, 47, 50, 51]. It is noteworthy that not only specific activities of the venom change but also its lethality might vary according to the snake's age even in a species with specialized diets such as *B. alternatus*, that feeds only on mammals throughout its life [24, 47]. Compositional differences between the venom of neonates and adults were observed in different species of viperids, such as *Bothriechis schlegelii*, *B. lateralis*, *Crotalus durissus* ssp, *C. viridis viridis*, *Lachesis muta*, and *L. stenophrys*, as well as in the elapid species, *Naja kaouthia* [52–59]. Such widespread phenomenon highlights the importance of studying the ontogenetic shift in the venom of snakes, as this information can improve the treatment of snakebite patients and the antivenom production. Venom intraspecific variations have been described to result in different reactions *in vitro* and *in vivo* (experiments and clinical reports) between the venoms and their specific antivenoms [27, 32, 33, 46, 49, 60].

Whilst extensively studied, intraspecific variability still needs more elucidation, as it could have an impact in snakebite treatment [36, 61]. In this work, we analyzed the venom of captive-born Brazilian lancehead at different life stages in controlled environment.

## Material and methods

### Venoms and animals

Captive-born Brazilian lancehead were milked in the Laboratory of Herpetology at Instituto Butantan, according to the laboratory's protocol [62], centrifuged at 2,500 x *g* for 15 min to remove debris, lyophilized, and stored at −20˚C until the analyses. Protein concentration of the venoms was determined according to the Bradford method, using Bio-Rad Protein Assay reagent (5000006) and bovine serum albumin (Sigma, A2153) as standard, following the manufacturer's recommendations. Table 1 shows the groups by ages, number of specimens, average size (snout-vent length–SVL in cm), protein concentration, and locality of the snakes or their parents; in S1 Table, individual information is listed.

All experiments involving snakes and mice were approved by the Ethical Committee for the Use of Animals of Instituto Butantan (CEUAIB), São Paulo, Brazil, protocol identification numbers 9238080117, 2540200919, and 7967310720, and were certified in agreement with the Ethical Principles in Animal Research adopted by the Brazilian Council of Animal Experimentation Control (CONCEA).

### Compositional analysis

**One-dimensional electrophoresis (1-DE).** Twenty micrograms of venom samples were homogenized with a sample buffer in the presence of β-mercaptoethanol (Calbiochem, 444203). 1-DE was carried out in 15% gels, according to the method described by Laemmli [63], and then gels were stained with Coomassie Blue G-250 (Sigma. B0770) according to manufacturer's recommendations.

**Reverse-phase high performance liquid chromatography (RP-HPLC).** Five hundred micrograms of lyophilized individual and pooled venoms were dissolved in 200 μL of 0.1% trifluoroacetic acid (TFA (Sigma, 302031); solution A), centrifuged at 13,000 x *g* for 15 min, and separated by RP-HPLC using a Teknokroma Europa Protein 300 C18 column (0.46 cm × 25 cm, 5 mm particle size, 300 Å pore size) and an Äkta Purifier system (GE Healthcare).

**Table 1. Data of age, size, and localization where the snakes or their parents came from.**

| Group | Age | N | | SVL (cm ± SD) | | mg protein/mg venom | | Locality |
|---|---|---|---|---|---|---|---|---|
| | | Male | Female | Male | Female | Male | Female | |
| BmN | Neonate | 18 | 16 | 26.9 ± 1.1 | 27.9 ± 1.0 | 0.65 ± 0.01 | 0.67 ± 0.01 | Araraquara–SP |
| | (24 days) | | | | | | | |
| Bm<1 | > 1 year-old | 18 | 15 | 35.2 ± 3.5 | 35.4 ± 2.8 | 0.65 ± 0.03 | 0.78 ± 0.003 | Patos de Minas–MG |
| | (5 months) | | | | | | | Brotas–SP |
| Bm1 | 1 year-old | 6 | 3 | 50.9 ± 3.3 | 51.2 ± 1.6 | 0.73 ± 0.02 | 0.73 ± 0.01 | Chapada dos Guimarães–MT |
| | | | | | | | | Gaúcha do Norte–MT |
| Bm2 | 2 years-old | 8 | 8 | 64.3 ± 3.6 | 65.6 ± 8.4 | 0.80 ± 0.01 | 0.74 ± 0.01 | Gaúcha do Norte–MT |
| | | | | | | | | Palmas–TO |
| Bm3 | 3 years-old | 1 | 5 | 78.5 ± 0.0 | 96.5 ± 3.3 | 0.76 ± 0.02 | 0.82 ± 0.02 | Chapada dos Guimarães–MT |
| Bm>4 | > 4 years-old | 5 | 5 | 98.2 ± 3.8 | 112.3 ± 8.0 | 0.98 ± 0.03 | 1.31 ± 0.03 | Araraquara–SP |
| | | | | | | | | Chapada dos Guimarães–MT |
| | | | | | | | | Gaúcha do Norte–MT |
| | | | | | | | | Palmas–TO |

Group: the identification of species and age used throughout this work. Age: the age of the snakes when their venoms were milked. N: number of individuals (male and female). SVL (cm ± SD): the average size (snout-vent-length in cm), and standard deviation in each group. mg protein/mg venom: protein concentration in each pool used. Locality: the place where the individuals or their parents came from. Detailed information of each individual is listed in S1 Table.

Elution was carried out at 1 mL/min by applying a gradient toward solution B (95% acetonitrile (Sigma, 34851) containing 0.1% TFA), according to Gay *et al.* [64] with some modifications: 5% B for 5 min, 5–25% B for 10 min, 25–45% B for 60 min, 45–70% B for 10 min, 70–100% B for 10 min, and 100% B for 10 min.

**Mass spectrometry (LC-MS/MS).** Venom protein samples of 100 μg were dissolved in 50 μL of 50 mM ammonium bicarbonate (Sigma, 9830), followed by addition of 25 μL of 0.2% RapiGest (Waters), and incubation at 80˚C for 15 min. Samples were reduced with 5 mM dithiothreitol (Sigma, D9779) at 60˚C for 30 min and then alkylated in the dark with 10 mM iodoacetamide (Sigma, I1149) at room temperature for 30 min. Proteins were digested using trypsin (Promega, V5111) at a 1:100 (wt:wt) enzyme:protein ratio at 37˚C overnight. Digestion was stopped by addition of 10 μL of 5% TFA (Sigma, T6508) and incubation at 37˚C for 90 min. Samples were filtered through 0.22 μm Millex-GV filters (EMD Millipore) into glass vials. The final protein concentration was approximately 2 μg/μL. Mass spectrometry experiments of venom digests were performed on a Synapt G2 HDMS (Waters) mass spectrometer coupled to a nanoAcquity UPLC system (Waters). Approximately 10 μg of each peptide mixture was loaded online for 5 min at a flow rate of 8 μL/min of phase A (0.1% formic acid (Sigma, F0507)) using a Symmetry C18 trapping column (5 μm particles, 180 μm x 20 mm length; Waters). The mixture of trapped peptides was subsequently separated by elution with a gradient of 7–35% of phase B (0.1% formic acid in acetonitrile (Sigma, 34851)) through a BEH 130 C18 column (1.7 μm particles, 75 x 150 mm; Waters) in 90 min at 325 nL/min. Data were acquired in the data-independent mode HDMS$^E$ with ion mobility separation in the m/z range of 50–2000 and resolution mode. Collision energies were alternated between 4 eV and a ramp of 19–48 eV for precursor ions and fragment ions, respectively, using scan times of 1.0 s. The ESI source was operated in positive mode with a capillary voltage of 3.1 kV, block temperature of 100˚C, and cone voltage of 40 V. For lock mass correction, a [Glu1]-Fibrinopeptide B solution (500 fmol/mL in 50% acetonitrile, 0.1% formic acid; Peptide 2.0) was infused through the reference sprayer at 500 nL/min and sampled every 60 s. Venom samples were analyzed in technical duplicates. Mass spectrometry raw data were processed in ProteinLynx Global Server 3.0.3 (Waters) platform using a low energy threshold of 750 counts and an elevated energy threshold of 50 counts. Database searches were performed against *Bothrops* sequences from UniprotKB/Swissprot (www.uniprot.org; unreviewed; 3,555 sequences; downloaded on August 29th, 2019). The following search parameters were used: automatic tolerances for precursor and fragment ions, carbamidomethylation of cysteine residues as fixed modification, oxidation of methionine and N-terminal acetylation as variable modifications, and trypsin digestion with up to two missed cleavage sites allowed. Protein identifications were considered with a minimum of one fragment ion per peptide, five fragment ions per protein, two peptides per protein, and a false discovery identification rate set to 1%, estimated by simultaneous search against a reversed database [65].

Label-free quantification was performed in Progenesis QI for proteomics (NonLinear Dynamics, Newcastle, UK) as previously reported [66]. Briefly, the raw files were loaded in the software and a reference run for the replicates was automatically chosen. Precursor ion retention times were processed for alignment, peak picking and normalized to the reference run with default parameters. Relative quantification was carried out by the comparison of peptide ion abundances, which were calculated as the sum of the areas under the isotope boundaries.

## Functional analysis

**Collagenolytic activity.** Collagenolytic activity over azocoll was determined according to Váchová and Moravcová [67] and modified by Antunes *et al.* [49]. Venoms (6.25 μg) were

incubated with 50 µL of a 5 mg/mL azocoll (Merck, 194933-5GM) solution, both diluted in Tyrode buffer (137 mM NaCl (Merck, 1064041000), 2.7 mM KCl (ECIBRA, ACS1233), 3 mM NaH$_2$PO$_4$ (Sigma, S-9638), 10 mM HEPES (Sigma, H3375), 5.6 mM dextrose (Sigma, D9434-500G), 1 mM MgCl$_2$ (Sigma, M8266), 2 mM CaCl$_2$ (Sigma, C106), pH 7.4) for 1 h in constant shake, at 37°C. The samples were centrifuged for 3 min at 5,000 x *g* and the absorbance of the supernatants (200 µL) was measured at 540 nm in a SpectraMax i3 microplate reader (Molecular Devices). One unit of activity was determined as the amount of venom that induces an increase of 0.003 units of absorbance. Specific activity was expressed as U/min/mg of venom.

**Caseinolytic activity.**　Caseinolytic activity was determined as described by Menezes *et al.* [42]. Briefly, 10 µL of venom solution (1 mg/mL) and 500 µL of a solution of 2% N,N-dimethylated casein (Sigma, C9801), both solubilized in same buffer, were added to 490 µL of buffer (100 mM Tris (Sigma, T-8524), 10 mM CaCl$_2$, pH 8.8). The mixture was incubated for 30 min at 37°C. The reaction was stopped by adding 1 mL of 5% trichloroacetic acid (TCA; Sigma, T6399). The sample was then incubated for 10 min in ice bath, centrifuged at 14,000 x *g* at 4°C for 15 min, and absorbance of the supernatants was performed in the plate reader SpectraMax i3 (Molecular Devices) using a wavelength of 280 nm. Specific activity was expressed as U/min/mg.

**Phospholipase A$_2$ activity.**　Phospholipase A$_2$ activity was assayed based on the method described by Holzer and Mackessy [68] using the synthetic substrate 4-nitro-3-octanoyloxy-benzoic acid (NOBA; Enzo, BML-ST506-0250). Briefly, 20 µL of venom (1 mg/mL in 154 mM NaCl) were added to 200 µL of buffer (10 mM Tris, 10 mM CaCl$_2$, and 100 mM NaCl, pH 8.0) and 20 µL of H$_2$O in a 96-well microplate. Then, 20 µL of substrate (NOBA) (4.16 mM in acetonitrile) were added, to a final concentration of 0.32 nM. The mixture was homogenized and incubated for up to 60 min at 37°C, and the absorbance at 425 nm was measured in the plate reader SpectraMax i3 (Molecular Devices). A standard curve of absorbance as a function of chromophore (3-hydroxy-4-nitrobenzoic acid) concentration showed that a change in absorbance of 0.1 U was equivalent to 25.8 nmol of chromophore release [68]. The PLA$_2$ specific activity was expressed as U/min/mg of venom.

**L-amino acid activity.**　The microplate assay for LAAO activity was conducted as described by Kishimoto and Takahashi [69] with slight modifications. Ten microliters of venom (1 mg/mL in 154 mM NaCl) were added to the reaction mixture (50 mM of Tris, pH 8.0, 250 mM L-methionine (Sigma, M9265), 0.8 U/mL horseradish peroxidase (Sigma, P8125-5KU), and 2 mM of orthophenylenediamine (Merck, 1072430050). The reaction was incubated for 1 h at 37°C and stopped by adding 50 µL of 2 M H$_2$SO$_4$ (Anidrol, PAP.A-8718). Absorbance was determined at 492 nm in a SpectraMax i3 microplate reader (Molecular Devices). Hydrogen peroxidase standards were used, and the linear regression data were calculated with Microsoft Excel software. Specific activity was expressed as µM/min/mg of venom.

**Minimum coagulant dose (MCD).**　The coagulant activity of the venom pools was assessed in citrated plasma according to Theakston and Reid [70]. Briefly, 100 µL of plasma were incubated at 37°C for 60 s. After the incubation, 50 µL of various doses of venom (15.625 −1,000 µg) were mixed to the plasma and clotting times were measured in a coagulometer (MaxCoag, MEDMAX). The Minimum Coagulant Dose (MCD) was defined as the minimum amount of venom that induced coagulation of plasma in 60 s at 37°C.

**Hemorrhagic activity.**　The hemorrhagic activity of BmN and Bm>4 venoms was determined according to Nadur-Andrade *et al.* [71] and Gutiérrez *et al.* [72]. To reduce the number of animals used in this test, doses were calculated only from the most extreme groups (neonate and adult, male and female venoms were mixed). Groups of male Swiss mice weighing 18–22 g (n = 3 for each venom and for control) were injected intradermally in the abdomen with 20 µg of venom (100 µL of solution/mouse) dissolved in sterilized 0.15 M NaCl; control group was injected with only 100 µL of sterilized 0.15 M NaCl. After 3 h, the animals were euthanized in a

$CO_2$ chamber, and the abdominal skin was removed, photographed and digitalized. The hemorrhagic area was obtained using the ImageJ software version 1.53a.

**Lethal dose 50%.** The Median Lethal Dose ($LD_{50}$) is the dose required to kill half the members of a tested population. To reduce the number of animals used in this test, doses were calculated only from the most extreme groups (neonate and adult, in both groups male and female venoms were mixed). The $LD_{50}$ of *B. moojeni* venoms was determined in groups of male Swiss mice weighing 18–22 g (n = 5 for each dose), injected intraperitoneally with five doses (500 μL of solution containing 57, 88, 136, 211 or 327 μg/mouse) of each venom (BmN and Bm>4) dissolved in sterilized 0.15 M NaCl. The number of deaths were recorded during 48 h and the venom's $LD_{50}$ was calculated by probit analysis [73].

### Immunointeraction *B. moojeni* venom *vs.* antibothropic serum

**Western blotting.** Venom samples (20 μg) separated by 1-DE were electro-transferred in a semi-dry system (Trans-Blot Turbo Transfer System, Bio-Rad) onto a PVDF membrane, previously equilibrated in transfer buffer (25 mM Tris, 192 mM glycine (Anidrol, PAP.A-0999), 20% ethanol (Merck, 1117274000)). It was used a constant current of 1.0 A and voltage up to 25 V for 35 min. Thereafter, the membrane was blocked with TBS-milk (Tris-buffered 0.15 M NaCl containing 5% non-fat milk and 0.1% Tween 20 (Sigma, P1379)) overnight at 4˚C. The membrane was washed twice with wash buffer (10 mM Tris, 150 mM NaCl, 0.1% Tween 20; pH 7.5) and incubated with 1:5,000 commercial antibothropic serum for 2 h at 4˚C. After 3 washes using wash buffer, the membrane was exposed to 1:10,000 peroxidase-labelled anti-horse IgG (Sigma, A6917) for 2 h at 4˚C. The membrane was washed 3 times again and reaction was developed with diaminobenzidine (Sigma, D8001) and $H_2O_2$ (Merck, 1072090250) [74]. The commercial antibothropic serum is produced at Instituto Butantan by hyperimmunization of horses using a mixture of *Bothrops* species venoms: *B. jararaca* (50%), *B. alternatus* (12.5%), *B. jararacussu* (12.5%), *B. moojeni* (12.5%), and *B. neuwiedi* complex (12.5%).

### Statistical analysis

Results are expressed as mean ± SD of triplicates. We compared the venoms' activities from snakes of the same sex of different ages by one-way ANOVA with Tukey as *post-hoc* test. Male and female venoms of the same age, as well as hemorrhagic activity and $LD_{50}$, were analyzed by Student's *t*-test using GraphPad Prism 7.03 software; $p < 0.05$ was considered significant. Statistical significance is listed in S2 Table.

## Results

### Compositional analysis

An ontogenetic variation in the composition of the venoms is notable on 1-DE profiles of males and females *B. moojeni* at different ages (Fig 2). According to previous works [31, 33, 75–79], protein families were assigned. Therefore, a single band of ~150 kDa, absent in both neonate venoms (BmN), may correspond to phosphodiesterase (PDE). Likewise, a band between 20 and 25 kDa, possibly P-I SVMP, nerve growth factor (NGF), and/or cysteine-rich secretory protein (CRISP), appears in the venoms of individuals from Bm1 onward. The bands between 50 and 75 kDa that decrease in intensity are most likely P-III SVMPs and LAAOs. The two bands between 25 and 37 kDa that are absent only in the venom of male BmN, female BmN and Bm1 are located in the region usually assigned to SVSP, CRISP, and P-I and P-II SVMP, where other bands got more intense in the pools of older snakes. Additionally, in the

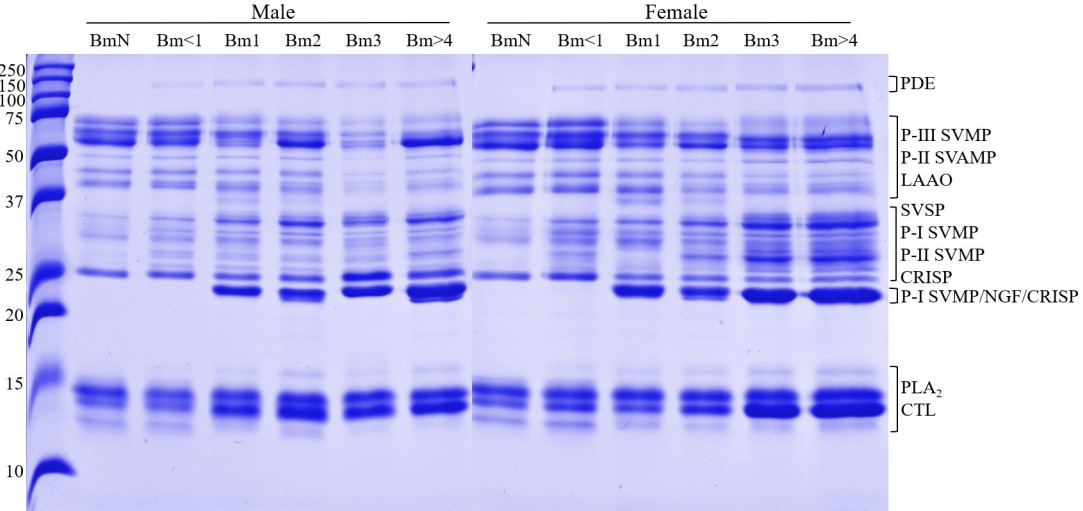

**Fig 2. One-dimensional electrophoresis (1-DE, 15%) profiles of male and female *B. moojeni* at different ages.** The samples (20 μg) were applied to the gel under reducing conditions (β-mercaptoethanol) and the protein bands were stained with Coomassie Blue G-250, according to the manufacturer's recommendation. Molecular weight markers (Dual Color Precision Plus, BioRad) are shown on the left side. PDE: phosphodiesterase; SVMP: snake venom metalloprotease; LAAO: L-amino acid oxidase; SVSP: snake venom serine protease; CRISP: cysteine-rich secretory protein; NGF: nerve growth factor; PLA₂: phospholipase A₂; CTL: C-type lectin.

area comprising bands of 10–15 kDa are PLA$_2$s and CTLs, three bands show increased intensity in older pools and one shows the opposite.

We found some quantitative differences in the chromatographic profiles of the venoms studied (Figs 3 and 4). Protein families were assigned to each region in the chromatograms according to a previous report [11]. An intense peak before 20 min (disintegrin–DIS) was observed in all venoms. This peak was highest in female BmN and Bm<1 (Fig 4), and in Bm1 of males (Fig 3), while in the other venom samples it remained similar. After this initial peak, some weak peaks (also DIS) appeared near 20 min, more intense and diversified in the neonate groups and decreased in both aspects in older groups. The peaks found between 40 and 50 min (non-catalytic PLA$_2$) in the male chromatograms gradually increased until Bm1 and remained in Bm2. Bm3 venom showed a decrease in these peaks and Bm>4 remained at a similar intensity in comparison to Bm2. In females, these peaks increased only in Bm3 and Bm>4. Only male Bm2 presented three peaks in this interval.

An increase in variability was observed in proteins eluted between 50 and 70 min (SVSP, P-I SVMP, and D-49 PLA$_2$), mainly in males Bm1 and Bm2. However, in Bm3 the peak intensity decreased and the pattern seemed to be maintained in Bm>4. In the final region of the chromatogram (predominantly P-III SVMP), it was also possible to observe an increase in variability and, in Bm2 and Bm>4, the appearing of a second high intensity peak (marked with * in Figs 3 and 4), which was not observed in Bm3. In fact, there was a noticeable loss of variability in this region in Bm3 venoms. It was also noticeable that, in general, the profiles of females' venoms were more conserved at different ages when compared to that of males.

By LC-MS/MS analysis of the twelve venoms used in this work, the following protein families were obtained: bradykinin-potentiating peptide (BPP), calmodulin, CRISP, CTL, DIS, flavin monoamine oxidase (FMAO), glutaminyl-peptide cyclotransferase (GPC), hyaluronidase (HYA), Kunitz-type serine protease inhibitor (KUNI), LAAO, leucine zipper tumor suppressor (LZTS), NGF, nucleotidase (NUC), PDE, SVMP, PLA$_2$, phospholipase B (PLB), phospholipase A$_2$ inhibitor (PLI), SVSP, TNF receptor-associated factor 6 (TRAF6), vascular endothelial growth factor

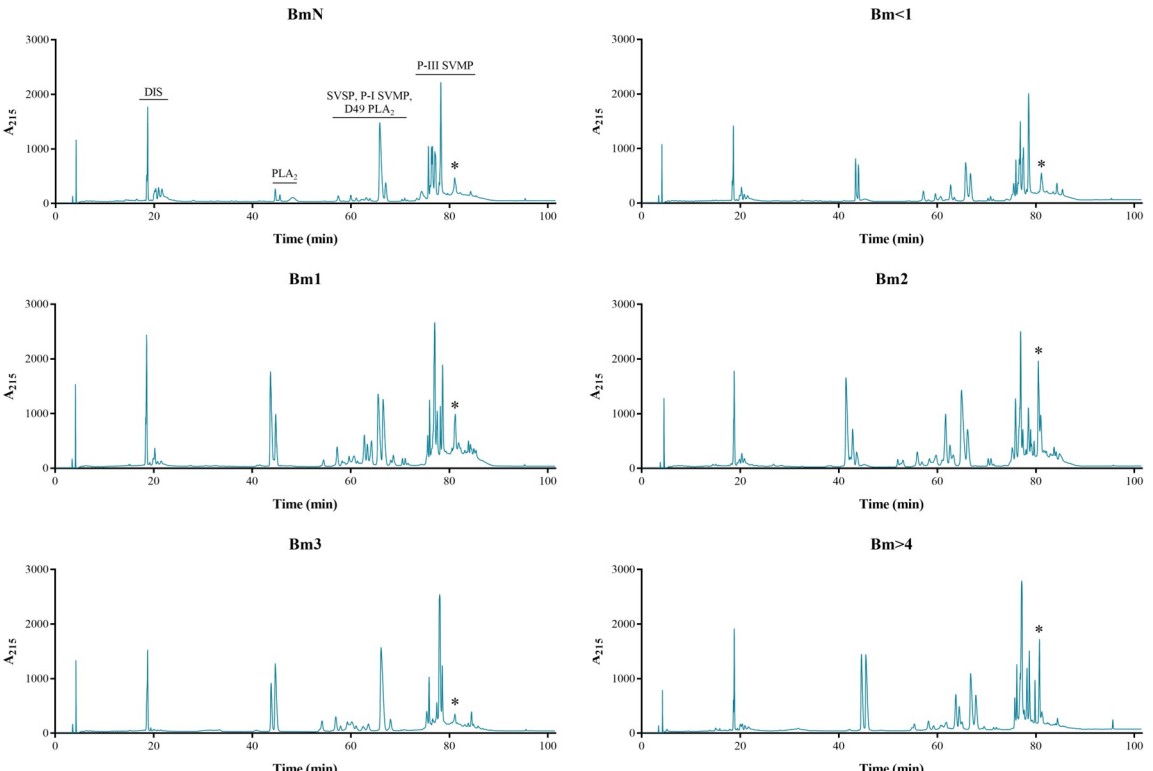

**Fig 3. Venom profile of male *B. moojeni* at different ages by RP-HPLC.** Samples of 500 µg of lyophilized venom were dissolved in 0.1% trifluoroacetic acid (TFA; solution A) and submitted to RP-HPLC in a C18 column. Elution was carried out at 1 mL/min by applying a solution B (95% acetonitrile containing 0.1% TFA) gradient: 5% for 5 min; 5–25% for 10 min, 25–45% for 60 min, 45–70% for 10 min, 70–100% for 10 min, and 100% for 10 min. Absorbance was measured at 215 nm. *: marked peak that may be related to collagenolytic activity. DIS: disintegrin; PLA₂: phospholipase A₂; SVSP: snake venom serine protease; SVMP: snake venom metalloprotease.

(VEGF), and waprin (WAP). The venoms presented all protein families listed, except for male Bm3 and female Bm>4 that did not show any S-49 PLA₂ (S3 and S4 Tables and Fig 5). The protein families BPP, calmodulin, DIS, FMAO, GPC, KUNI, NGF, PDE, PLB, PLI, TRAF6, and WAP were grouped into "Others", considering all venoms presented < 1% of each family.

Both male and female of the group BmN presented the highest abundance of CRISP, SVMP, and VEGF; CTL was found more abundant in Bm3 of both sexes; SVSP and HYA were higher in both Bm1 venoms. Higher abundance of LAAO was observed in the venoms of male Bm<1 and female Bm>4; LZTS was higher in males Bm3 and females Bm<1; as for NUC, higher abundance was observed in males BmN and females Bm>4; and finally, males Bm>4 and females Bm3 presented the highest abundance of PLA₂. Lower relative abundance of HYA, LAAO, and overall SVMP were observed in both males and females Bm3 venoms. In the venoms of males Bm>4 and females Bm3 were identified low abundance of CRISP; in the case of CTL males Bm<1 and females BmN were lower; males Bm>4 and females BmN presented the lowest abundance of LZTS; NUC was lower in the venoms of males Bm2 and females BmN; PLA₂ and SVSP were lower in males BmN, PLA₂ in females Bm1 and SVSP in females Bm3; and finally, males Bm>4 and females Bm3 presented low abundance of VEGF. Analyzing each class of SVMP, we obtained the following: higher amounts of P-I were present in both BmN venoms and lower in Bm 3; P-II was higher in males and females Bm<1 venoms and lower in males Bm>4 and females Bm3; and male Bm<1 also had the greatest concentration

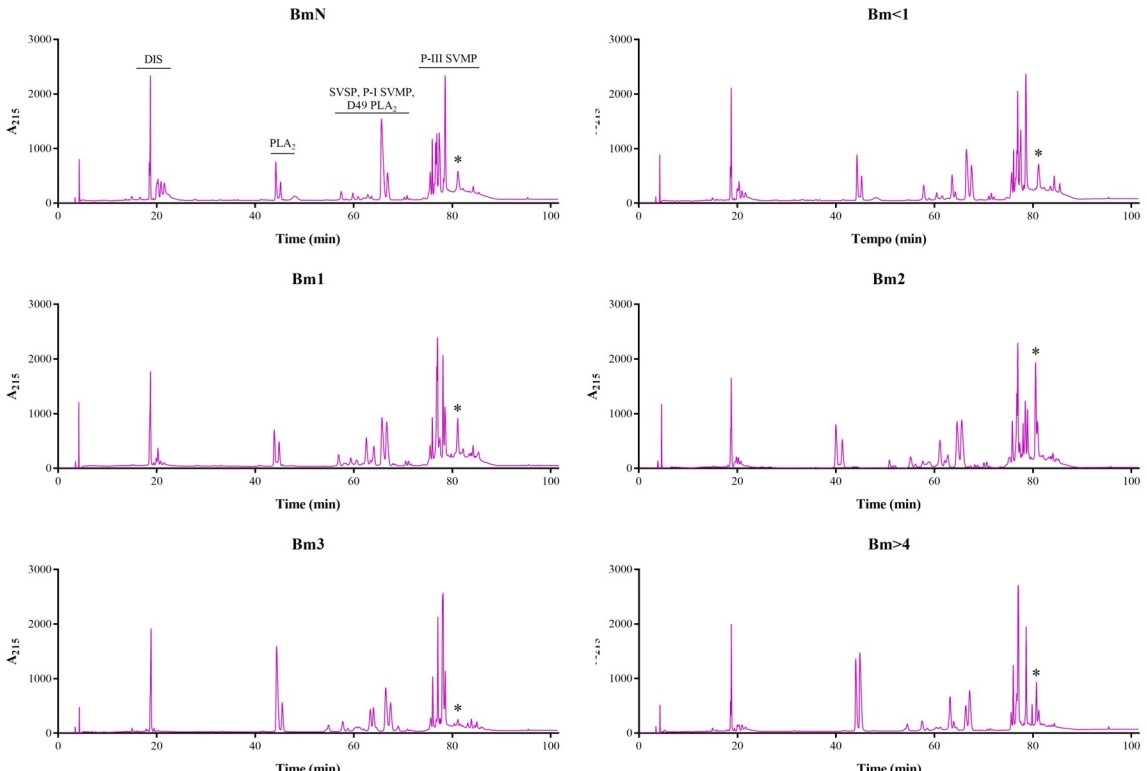

**Fig 4. Venom profile of female *B. moojeni* at different ages by RP-HPLC.** Samples of 500 μg of lyophilized venom were dissolved in 0.1% trifluoroacetic acid (TFA; solution A) and submitted to RP-HPLC in a C18 column. Elution was carried out at 1 mL/min by applying a solution B (95% acetonitrile containing 0.1% TFA) gradient: 5% for 5 min; 5–25% for 10 min, 25–45% for 60 min, 45–70% for 10 min, 70–100% for 10 min, and 100% for 10 min. Absorbance was measured at 215 nm. *: marked peak that may be related to collagenolytic activity. DIS: disintegrin; PLA$_2$: phospholipase A$_2$; SVSP: snake venom serine protease; SVMP: snake venom metalloprotease.

of P-III, along with females BmN, while males and females Bm3 showed the lowest quantity of this class (Table 2). Altogether, it was possible to observe increasing quantities of PLA$_2$ and a certain decrease of VEGF, P-I and P-III SVMP in both genders, as well as a decrease of P-II SVMP in females.

## Functional analysis

As one of the main effects caused by bothropic venom is the proteolytic activity, we assessed the effect of the venoms over casein and collagen. Increasing collagenolytic activity was observed according to the snakes age, except for Bm3, which showed an activity similar to that of Bm1 (Fig 6A). The most accentuated differences between males and females were seen in BmN, in which the activity of females was greater; and in Bm>4, in which the activity of males was higher than that of females, which remained the same as females Bm2.

Males' venoms showed an increase in caseinolytic activity according to age, but the same was not observed in females' venom, in which there was an increase in the activity until Bm2 and a decrease in the last two ages (Fig 6B). In addition, females' venoms activity were higher at all ages than males', except for Bm>4, when males' venoms had greater activity.

The PLA$_2$ activity (Fig 6C) of the venoms did not appear to be related to the age of the snakes, since in males and females this activity had no tendency to increase or decrease. It was not possible to establish whether the gender influenced this activity, considering that in

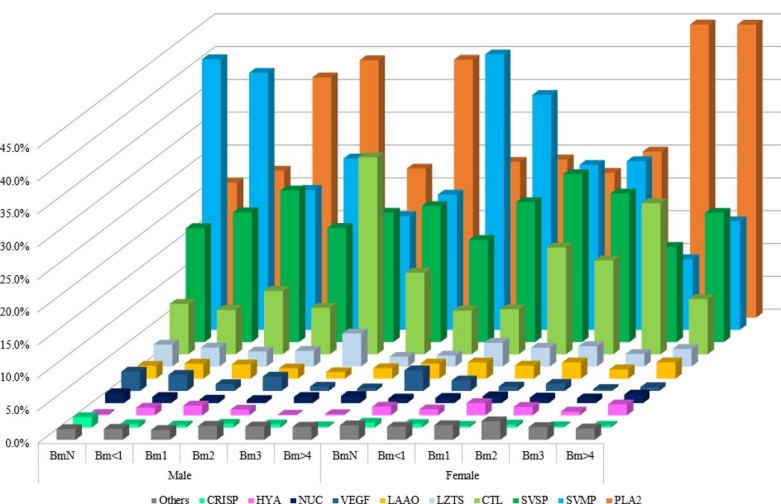

**Fig 5. Overview of protein families' abundances in male and female *B. moojeni* venoms at different ages.** All venom pools were subjected to LC-MS/MS, and proteins were identified, quantified, and assigned to each protein family expressed as percentage. CRISP: cystein-rich secretory protein; HYA: hyaluronidase; NUC: nucleotidase; VEGF: vascular endothelial growth factor; LAAO: L-amino acid oxidase; LZTS: leucine zipper tumor suppressor; CTL: C-type lectin; SVSP: snake venom serine protease; SVMP: snake venom metalloprotease; PLA$_2$: phospholipase A$_2$. Others: included families that accounted for < 1% in all venoms: bradykinin-potentiating peptide, calmodulin, disintegrin, flavin monoamine oxidase, glutaminyl-peptide cyclotransferase, Kunitz-type serine protease inhibitor, nerve growth factor, phosphodiesterase, phospholipase B, phospholipase A$_2$ inhibitor, TNF receptor-associated factor 6, and waprin.

females Bm2 and Bm>4 the PLA$_2$ activity was higher than in males of the same age, while in Bm3 it was lower. However, the PLA$_2$ activity of females appeared to be more stable than that of males.

In general, LAAO activity (Fig 6D) did not change significantly, however in Bm>4, mainly in females, there was a considerable increase in relation to other ages. In addition, females appeared to have greater activity than males, except in intermediate ages (Bm2 and Bm3), in which there were no significant differences.

The MCD proved to be stable during the animals' first year of life, with a large increase in Bm2 in both sexes (Fig 6E). In females, the coagulant activity seemed to gradually decreased in the last three groups, while in males, it decreased in Bm3 and increased to an intermediate value between Bm2 and Bm3, and in Bm>4. Although statistical difference was observed only between BmN and Bm2, males' venoms were more coagulant than that of females in all ages, except for Bm3.

Hemorrhagic halos ranged between 0.78–1.79 cm$^2$ (mean: 1.17 ± 0.55 cm$^2$) and 1.58–1.98 cm$^2$ (mean: 1.80 ± 0.20 cm$^2$) after injection of 20 μg of neonates' and adults' venoms (males and females mixed), respectively. Although the values obtained were not statistically different considering the standard deviation (Fig 7A), Bm>4 produced hemorrhagic halos more intense than those of BmN (data not shown).

As for LD$_{50}$, despite the variations observed throughout the other tests, BmN and Bm>4 did not show significant difference: 113.02 μg/animal (confidence interval: 83.12–161.04 μg/mouse) and 111.75 μg/animal (confidence interval: 72.14–153.92 μg/animal), respectively (Fig 7B). After the injection of the venoms, mice were observed for the first 6 h. During this time, we noticed that in higher doses, mice injected with Bm>4 venom seemed to die faster than those injected with BmN venom (Fig 7C and 7D). After this time and with the two lower doses (57 and 88 μg/animal), no mice died.

**Table 2. Relative abundance of the protein families obtained in each venom by LC-MS/MS.**

| | Male | | | | | | Female | | | | | |
|---|---|---|---|---|---|---|---|---|---|---|---|---|
| | BmN | Bm<1 | Bm1 | Bm2 | Bm3 | Bm>4 | BmN | Bm<1 | Bm1 | Bm2 | Bm3 | Bm>4 |
| **BPP** | 0.04% | 0.12% | 0.06% | 0.03% | 0.03% | 0.02% | 0.30% | 0.13% | 0.18% | 0.09% | 0.02% | 0.01% |
| **Calmodulin** | 0.11% | 0.04% | 0.05% | 0.09% | 0.09% | 0.01% | 0.15% | 0.10% | 0.12% | 0.11% | 0.16% | 0.13% |
| **CRISP** | 1.6% | 0.5% | 0.3% | 0.6% | 0.5% | 0.2% | 0.7% | 0.5% | 0.3% | 0.4% | 0.2% | 0.3% |
| **CTL** | 7.7% | 6.8% | 9.6% | 7.1% | 30.0% | 12.4% | 6.7% | 6.8% | 16.3% | 14.3% | 23.0% | 8.4% |
| **DIS** | 0.0003% | 0.0008% | 0.0018% | 0.0831% | 0.0122% | 0.0006% | 0.0008% | 0.0075% | 0.0309% | 0.0039% | 0.0136% | 0.0035% |
| **FMAO** | 0.14% | 0.07% | 0.03% | 0.03% | 0.07% | 0.04% | 0.09% | 0.06% | 0.07% | 0.06% | 0.10% | 0.04% |
| **GPC** | 0.31% | 0.22% | 0.28% | 0.56% | 0.57% | 0.63% | 0.21% | 0.30% | 0.37% | 0.49% | 0.45% | 0.48% |
| **HYA** | 0.2% | 1.2% | 1.5% | 0.9% | 0.1% | 0.2% | 1.3% | 0.9% | 1.9% | 1.3% | 0.6% | 1.7% |
| **KUNI** | 0.01% | 0.08% | 0.14% | 0.08% | 0.02% | 0.03% | 0.07% | 0.09% | 0.07% | 0.33% | 0.18% | 0.14% |
| **LAAO** | 2.0% | 2.3% | 2.2% | 1.6% | 1.0% | 1.6% | 2.3% | 2.5% | 2.0% | 2.5% | 1.4% | 2.5% |
| **LZTS** | 3.3% | 2.9% | 2.3% | 2.4% | 5.0% | 1.5% | 1.7% | 3.6% | 2.9% | 3.1% | 1.9% | 2.7% |
| **NGF** | 0.01% | 0.01% | 0.01% | 0.10% | 0.07% | 0.08% | 0.02% | 0.01% | 0.03% | 0.09% | 0.03% | 0.09% |
| **NUC** | 1.5% | 1.0% | 0.5% | 0.5% | 0.9% | 1.1% | 0.7% | 0.8% | 1.1% | 0.8% | 0.7% | 1.3% |
| **PDE** | 0.53% | 0.53% | 0.49% | 0.55% | 0.59% | 0.61% | 0.59% | 0.63% | 0.80% | 0.98% | 0.50% | 0.39% |
| **P-I SVMP** | 15.0% | 5.7% | 3.3% | 3.3% | 1.3% | 3.2% | 7.4% | 5.2% | 3.9% | 5.0% | 1.5% | 3.4% |
| **P-II SVMP** | 5.6% | 6.0% | 3.6% | 3.6% | 5.3% | 3.5% | 5.8% | 6.3% | 3.5% | 3.2% | 2.5% | 3.0% |
| **P-III SVMP** | 20.7% | 27.4% | 14.4% | 19.2% | 10.7% | 13.9% | 28.7% | 24.3% | 17.8% | 17.5% | 6.8% | 10.1% |
| **D-49 PLA2** | 8.9% | 6.1% | 6.9% | 10.7% | 3.7% | 14.8% | 8.2% | 7.0% | 4.8% | 3.4% | 10.9% | 4.1% |
| **K-49 PLA2** | 11.7% | 16.3% | 29.7% | 28.5% | 19.0% | 24.5% | 15.6% | 16.9% | 17.3% | 21.9% | 33.8% | 40.5% |
| **S-49 PLA2** | 0.0002% | 0.0229% | 0.0007% | 0.0040% | 0.0000% | 0.0002% | 0.0112% | 0.1258% | 0.0003% | 0.0012% | 0.0004% | 0.0000% |
| **PLB** | 0.32% | 0.39% | 0.35% | 0.33% | 0.53% | 0.35% | 0.24% | 0.37% | 0.50% | 0.49% | 0.45% | 0.31% |
| **PLI** | 0.1362% | 0.1473% | 0.0395% | 0.2108% | 0.0337% | 0.2014% | 0.4713% | 0.2335% | 0.0158% | 0.1050% | 0.0001% | 0.1098% |
| **SVSP** | 17.4% | 19.8% | 23.1% | 17.4% | 19.7% | 20.7% | 15.6% | 21.3% | 25.6% | 22.6% | 14.5% | 19.7% |
| **TRAF6** | 0.00004% | 0.00043% | 0.00029% | 0.00235% | 0.00016% | 0.00002% | 0.00083% | 0.00164% | 0.00203% | 0.00035% | 0.00009% | 0.00092% |
| **VEGF** | 2.9% | 2.4% | 1.0% | 2.2% | 0.6% | 0.4% | 3.1% | 1.6% | 0.7% | 1.1% | 0.3% | 0.6% |
| **WAP** | 0.03% | 0.06% | 0.03% | 0.06% | 0.01% | 0.02% | 0.06% | 0.04% | 0.04% | 0.07% | 0.03% | 0.02% |

Percentage of each protein family identified and quantified in male and female *B. moojeni* venoms at different ages. BPP: bradykinin-potentiating peptide; CRISP: cystein-rich secretory protein; CTL: C-type lectin; DIS: disintegrin; FMAO: flavin monoamine oxidase; GPC: glutaminyl-peptide cyclotransferase; HYA: hyaluronidase; KUNI: Kunitz-type serine protease inhibitor; LAAO: L-amino acid oxidase; LZTS: leucine zipper tumor suppressor; NGF: nerve growth factor; NUC: nucleotidase; PDE: phosphodiesterase; SVMP: snake venom metalloprotease; PLA$_2$: phospholipase A$_2$; PLB: phospholipase B; PLI: phospholipase A$_2$ inhibitor; SVSP: snake venom serine protease; TRAF6: TNF receptor-associated factor 6; VEGF: vascular endothelial growth factor; WAP: waprin.

## Immunointeraction

Western blotting showed that the venom proteins, in general, were recognized by the serum, since the revealed bands formed a profile remarkably similar to the gels (Fig 8). However, there was a shortage of bands in the region between 25 and 37 kDa. It was possible to observe that the recognition of a band in the region between 10−15 kDa decreased with age, the opposite of what was observed in the 1-DE, in which the band intensified in this region.

## Discussion

In this work, we observed ontogenetic shifts, especially related to coagulation, proteolytic, and, to some extent, hemorrhagic activities of *B. moojeni* venom. These differences may influence the outcome of a snakebite caused by a neonate, juvenile, or adult snake.

Regarding the composition, some clear differences were observed. Ontogenetic variations in 1-DE profile have been previously reported in other *Bothrops* species. As observed herein,

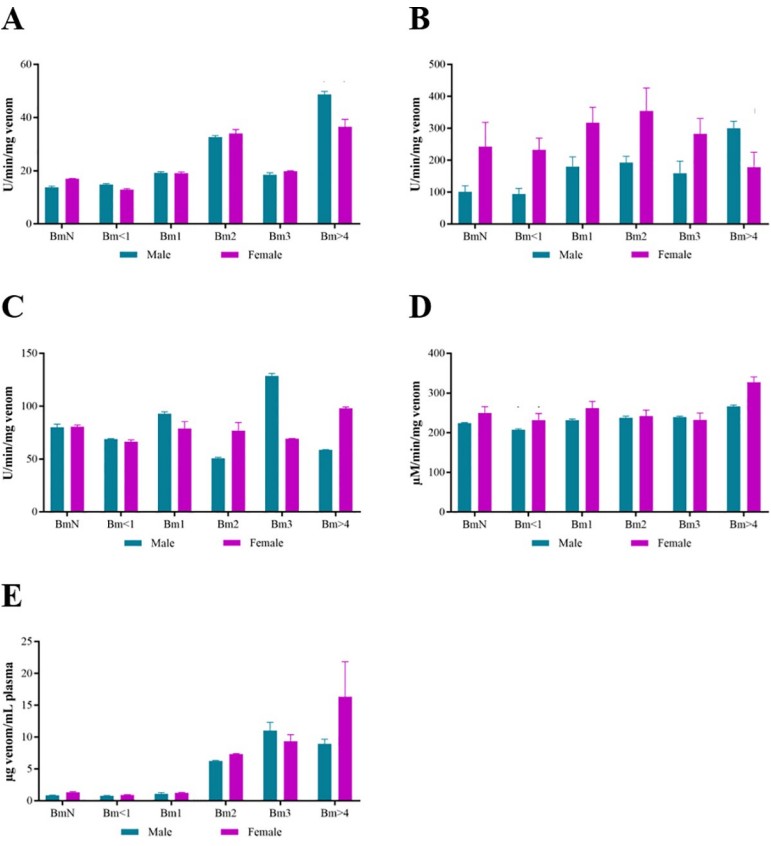

**Fig 6. *In vitro* activities of the venom of male and female *B. moojeni* at different ages.** Proteolytic activity over collagen (A) and casein (B), PLA$_2$ activity over the synthetic substrate 4-nitro-3-octanoyloxy-benzoic acid (C), LAAO activity with L-methionine (D), and coagulant activity over human plasma (E). Statistical data of all the groups is shown in S2 Table.

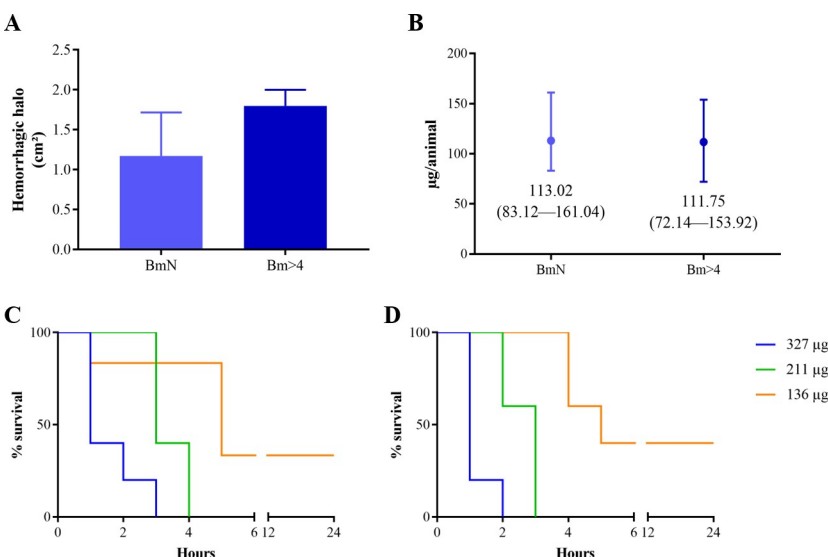

**Fig 7. *In vivo* assays of neonates and adults *B. moojeni* venom.** Hemorrhagic activity (A) was carried out by measuring the halos after intradermal injection of 20 µg venom. Median lethal dose (B) was determined by intraperitoneal injection of different doses of venom and the use of probit to obtain the values. The mice were observed for 6 consecutive hours and the number of deaths registered at each hour and 24 h after the injection of BmN (C) and Bm>4 (D) venoms.

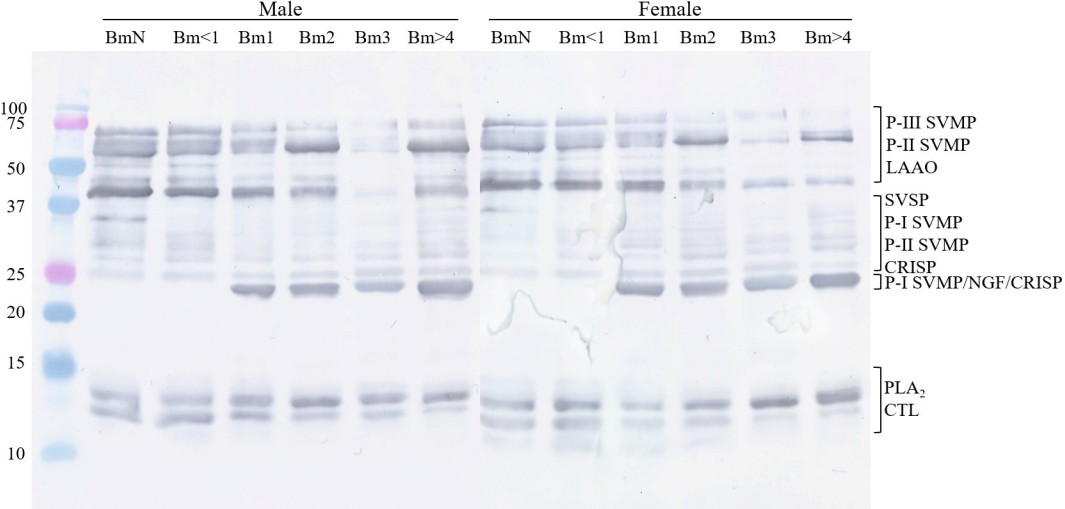

**Fig 8. Immunointeraction between *B. moojeni* venoms and commercial antibothropic serum.** The interaction of the proteins in *B. moojeni* venom with the antivenom produced at Instituto Butantan was assayed by western blotting. The samples were submitted to 1-DE, the proteins were electrotransferred onto a PVDF membrane, and incubated with the antibothropic serum. SVMP: snake venom metalloprotease; LAAO: L-amino acid oxidase; SVSP: snake venom serine protease; CRISP: cysteine-rich secretory protein; NGF: nerve growth factor; PLA$_2$: phospholipase A$_2$; CTL: C-type lectin.

some high molecular weight bands showed decreasing intensity and others were lost, as previously observed in the venoms of *B. jararaca* [48], *B. asper* [50], and *B. atrox* [50, 80]. López-Lozano and collaborators observed the appearing of a ~23 kDa band and the change of low molecular weight proteins in *B. atrox* venom [80]. In addition, the differences between venoms of neonates and adults observed herein were quite comparable to those obtained in *B. jararaca* [49]. Moreover, Guércio and colleagues [41] analyzed the venoms of *B. atrox* at three life stages and noticed that the venom of juveniles presented higher amounts of P-III SVMP, and more P-I and P-II SVMP in adults. Although we indeed obtained higher abundance of P-III SVMP at younger ages, the opposite was not observed with P-I and P-II SVMPs in our LC-MS/MS analysis (Fig 5 and Table 2). This might be caused by the different approaches utilized to study the venoms, considering it is known to cause differences in results [81, 82].

The protein composition analyzed by LC-MS/MS of the venoms showed evident ontogenetic differences, with differentiation of some protein families. Although there are some proteins missing in specific groups, there is no exclusive protein family present in any age or gender group, but there is a substantial quantitative variability of these families related to age. It is possible to observe a decrease of overall SVMP from Bm1 onward, while PLA$_2$ increases in both genders, earlier in males (Bm1) and later in females (Bm3). The astonishing relative abundance of PLA$_2$ obtained in some venoms in this work were previously reported in the adult venom of *B. asper* [83], although not usual. Curiously, some venoms presented great amounts of CTL ($>$ 10%), which is also not usual, but have been observed for *B. jararaca*, *B. alternatus*, and *B. atrox* [11, 84].

Interestingly, a leucine zipper tumor suppressor 1 (K4GPX1) was identified and considerably quantified (1.5–5.0%). According to the information available at UniProtKB (https://www.uniprot.org/uniprot/K4GPX1), this protein was also identified in *B. asper* venom. There is only one work related to the subject [85], where the authors sequenced genes in order to clarify some phylogenetic relationships within Squamata.

Comparing the RP-HPLC profile of the venoms used herein with their respective collagenolytic activity, a pattern could be observed (Figs 3, 4, and 6A). A similar occurrence was

reported for adult *B. jararaca* venoms by Galizio *et al.* [34] and for adult *B. moojeni* venoms by Aguiar *et al.* [86], in which some individuals showed close to no collagenolytic activity that was related to the absence of one to two peaks in the region comprising P-III SVMPs, which may indicate the presence of a collagen-specific protease.

As observed herein, higher proteolytic activity presented by adult venoms was reported by other authors [26, 87], including Furtado *et al.* studying *B. moojeni* [28]. Interestingly, caseinolytic activity among females increased until 2 years-old and decreased afterwards. In a previous work with *B. jararaca* venoms, the newborns presented lower activity than adults over collagen, but obtained similar values between the pools over casein [49]. Overall, our results showed that females were likely to have higher caseinolytic activity than males (except for Bm>4), corroborating previous studies in which variation according to gender was observed [42, 88]. Higher proteolytic activity in females' adult venoms might be related to the snakes' size. Female-biased sexual size dimorphism is present in *Bothrops* genus, so adult females are heavier, larger, and present a relatively bigger head compared to adult males [25, 88–90]. An ontogenetic shift in diet from ectotherm to endotherm organisms was observed for *B. moojeni* [25]. Despite the fact that this species keeps feeding on ectotherms for a long period of their development, larger animals, especially females, tend to feed mainly on mammal preys [24, 91].

In contrast to the proteolytic assay, LAAO and PLA$_2$ activities did not show such an evident ontogenetic shift. A significant difference between genders was found in all values of enzymatic activities in the older groups (Bm>4), in which males showed greater proteolytic activity, while females in LAAO and PLA$_2$ activities. However, it is difficult to affirm that such differences are related to sex, due to the inconsistency along different ages, especially in caseinolytic activity, where females showed much greater activity than males in all other ages, except Bm4. Sexual-related difference in LAAO activity of *B. moojeni* adult venom was observed elsewhere [33], but the females' venom pool showed higher activity than that of males'. On the other hand, when individual *B. moojeni* venoms were analyzed, no gender seemed to present higher or lower activity [86]. In contrast, PLA$_2$ activity in both aforementioned studies with *B. moojeni* venom [33, 86] was higher among males' venoms, and it was observed in this work only with Bm1 (no statistical difference) and Bm3. Snake venom PLA$_2$s from the Viperidae family can be divided into two groups: catalytically active (D-49) and enzymatically inactive (K-49, S-49, R-49, Q-49 and N-49) [79, 92, 93]. The presence of both types of PLA$_2$s in different proportions and the variability of catalytic efficiency (specific activity) could explain why the PLA$_2$ activity in the enzymatic assays did not reflect the increase of PLA$_2$ found in LC-MS/MS [94, 95].

Concerning the procoagulant effect of the venoms, in both genders the first three ages analyzed presented high coagulant activity upon plasma, whereas Bm2, Bm3, and Bm>4 showed low activity. This result corroborates previous works [49, 50, 80], in which adults presented lower activity compared to neonates/juveniles, a common characteristic among *Bothrops* genus. Additionally, at all ages (except Bm3) males had higher coagulant activity. Sexual-related differences in coagulant activity have been reported among *B. jararaca* juveniles [96] and adults [42], in addition to adult *B. atrox* [37], although another work with newborn and adult *B. jararaca* did not present variation according to sex [36]. Moreover, Aguiar *et al.* [86] studied adult *B. moojeni* venoms and, from the 10 females, five had lower coagulant activity upon plasma than three males, while two females presented similar MCD than one male. However, since the study analyzed individual venoms, it is hard to compare with the results obtained herein, as pools were utilized. As pointed out by Galizio *et al.* [34], individual venoms exert great influence over the pool, that does not necessarily reflect an average of all venoms.

So far, some compositional and specific activities presented ontogenetic-related variation, such as the proteolytic and coagulant activities, as well as SVMPs and PLA$_2$s abundances. It is also interesting to notice that the variation did not increase or decrease continuously. This may be influenced by the pools' compositions, since the venoms were milked from different snake specimens to compose the pool of each age instead of following the same animals during their development. Besides, the activities do not necessarily increase/decrease constantly as observed previously with other species [29, 51]. In addition, enzymes may not display the same catalytic potency and subtract specificity, and the synergistic action of the venom compounds may influence the outcome of a snakebite; thereby, *in vivo* tests are necessary to better understand the importance of such differences concerning snakes' evolutionary history (feeding and defense) and clinical importance (ophidian accidents). Furthermore, in order to reduce the number of mice in this study and considering that the focus is the ontogenetic variability, *in vivo* tests were performed using a mixture of male and female pools, thus composing a pool of neonates and adults.

Our results regarding hemorrhagic activity suggest that neonate venoms might be less hemorrhagic than of adults. This would be in accordance with previous works, where higher doses of newborn venoms are needed to induce hemorrhage [48, 49]. Likewise, LD$_{50}$ did not present a statistically significant difference, similar to the result found by Furtado *et al*. [28] and Andrade *et al*. [26]. Interestingly, the fact that Bm>4 seemed to have killed the mice faster, especially in the higher doses (Fig 7C and 7D) might indicate prey preference among adults. Studies comparing the LD$_{50}$ between *B. moojeni*, *B. jararaca*, and *B. alternatus* showed that the lethality of the venom of both *B. moojeni* and *B. jararaca*, species that undergo diet shift from juveniles to adults, are lower to mice and higher to frogs when they are juveniles, but when adults it is the opposite; in the case of *B. alternatus*, a species that feeds only on mammals and does not change its diet regardless of age, the LD$_{50}$ of juveniles and adults were virtually the same for either frogs or mice [26, 47].

To test the immunoreaction of the proteins present in the venoms with the antibothropic serum produced at Instituto Butantan, a western blotting assay was performed. Despite displaying a similar pattern to the 1-DE, some areas showed differences. The lower recognition of bands in the region between 25 and 37 kDa was previously observed with many *Bothrops* species [10]. Additionally, Amorim and collaborators [33] observed that *B. moojeni* females' venom were better recognized by the serum than the males' venom, a difference noticeable only with Bm1, Bm2, and Bm3 venoms, although subtle.

Considering how snake venom individual variability may affect the pool [34, 37], we need to highlight that the Bm3 venoms used in this work had a particular characteristic: they were all from individuals of the same litter and male Bm3 was composed by only one individual. Collagenolytic activity, for example, as a proteolytic activity, is expected to be higher in older snakes [28, 49, 51] but Bm3 venoms showed an activity similar to BmN, Bm<1, and Bm1. Furthermore, the chromatographic profiles obtained for Bm3 venoms were more similar to BmN than to any of the other venoms. Although the venoms of adult snakes are expected to be more proteolytic and less coagulant than newborn and juvenile venoms, previous works with individual adult venoms showed that some adults may present these two activities at levels close to those of younger snakes [34, 86]. The venoms used in this work were milked from snakes raised in captivity (neonates and juveniles were also born from individuals raised in captivity), under controlled environment conditions, and same feeding schedule; nonetheless, ontogenetic and sexual differences were still observed. Additionally, the animals we had available to collect venoms originated or were born from snakes from different locations. This might have influenced some results, as it is known that geographic location may cause venom variation [97–99], although a previous work by our group with individual *B. moojeni* did not notice

differences related to geographic origin [86] and, to the best of our knowledge, there are no previous studies regarding geographic variation of *B. moojeni* venom.

The reason behind intraspecific variability of venoms is still not completely clear. Studies comparing the venom gland transcriptomes and the venom proteomes were carried out to enlighten snake venom composition [84, 100–103]; these works showed that the abundance of genes in the venom gland and of proteins in the venom itself may match or not. Amazonas and colleagues [84] pointed out that "core function" loci, transcripts that encode multifunctional proteins, corresponds to toxins that undergo ontogenetic shift in *B. atrox* and suggested that they may be controlled by promoters related to hormones involved in ontogenetic development. Durban and colleagues suggested that microRNAs play an important role by modulating ontogenetic shift in venom composition in *Crotalus simus simus* [104], *C. simus*, *C. tzabcan*, and *C. culminatus* [105] through up- and down-regulation of genes.

## Conclusions

Ontogenetic variation has been extensively studied as an important evolutionary feature and as a medically relevant matter. In this work, we observed that from neonates to 1 year-old, the pools showed low collagenolytic and high procoagulant activities, and Bm2 and Bm>4 presented the opposite, characteristics commonly observed in adult venoms in other species. Although $LD_{50}$ was very similar between neonates and adults, when time of death was considered, we noticed that neonates' venom seemed to kill the mice slower. Considering that these animals are generalists, this result may indicate the prey preference inherent of this species, even when maintained in captivity since previous generations and fed mammals since they were born. The variation observed in this work, together with many others, highlight the importance of studying intraspecific variability for ecological, evolutionary, and medical purposes.

## Supporting information

**S1 Raw images. Original images of 1-DE and western blotting used in this work without any editing.**
(PDF)

**S1 Table. Individual information of the Brazilian lancehead used in this work.** Identification: code used to identify each individual. Group: the group at where each individual was assigned to compose the pools. Sex: ♂—male; ♀—female. Weight (g): the weight of each specimen when the venom was milked. SVL–Total length (cm): snout-vent length and total length of the animals when the samples were collected. Localization: geographic origin of the individual or the parents.
(XLSX)

**S2 Table. Statistical data of *in vitro* activities.** Statistical difference between ages of the same sex (one-way ANOVA) and between both sexes of the same age (Student's t-test). NS: no statistical difference.
(XLSX)

**S3 Table. Proteins identified and quantified in the venoms of males' and females' Brazilian lancehead at different ages.** Identification of the proteins in each of the twelve B. moojeni venoms through label-free quantification by high resolution LC-MS/MS considering the average intensity of the three most intense peptides of each identified protein.
(XLSX)

**S4 Table. Proteins identified by LC-MS/MS and their peptide sequences.** Complete data of the protein identification through label-free quantification by high resolution LC-MS/MS. (XLSX)

## Author Contributions

**Conceptualization:** Daniela Miki Hatakeyama, Karen de Morais-Zani, Anita Mitico Tanaka-Azevedo.

**Data curation:** Daniela Miki Hatakeyama, Lídia Jorge Tasima, Nathália da Costa Galizio, Caroline Serino-Silva, Caroline Fabri Bittencourt Rodrigues, Daniel Rodrigues Stuginski, Sávio Stefanini Sant'Anna, Kathleen Fernandes Grego, Alexandre Keiji Tashima, Erika Sayuri Nishiduka, Karen de Morais-Zani, Anita Mitico Tanaka-Azevedo.

**Formal analysis:** Daniela Miki Hatakeyama, Lídia Jorge Tasima, Nathália da Costa Galizio, Caroline Serino-Silva, Caroline Fabri Bittencourt Rodrigues, Alexandre Keiji Tashima, Erika Sayuri Nishiduka, Karen de Morais-Zani, Anita Mitico Tanaka-Azevedo.

**Funding acquisition:** Karen de Morais-Zani, Anita Mitico Tanaka-Azevedo.

**Investigation:** Daniela Miki Hatakeyama, Lídia Jorge Tasima, Nathália da Costa Galizio, Caroline Serino-Silva, Caroline Fabri Bittencourt Rodrigues, Alexandre Keiji Tashima, Erika Sayuri Nishiduka, Karen de Morais-Zani, Anita Mitico Tanaka-Azevedo.

**Methodology:** Daniela Miki Hatakeyama, Lídia Jorge Tasima, Alexandre Keiji Tashima, Erika Sayuri Nishiduka, Karen de Morais-Zani, Anita Mitico Tanaka-Azevedo.

**Project administration:** Daniela Miki Hatakeyama, Karen de Morais-Zani, Anita Mitico Tanaka-Azevedo.

**Resources:** Daniel Rodrigues Stuginski, Sávio Stefanini Sant'Anna, Kathleen Fernandes Grego, Karen de Morais-Zani, Anita Mitico Tanaka-Azevedo.

**Software:** Daniela Miki Hatakeyama, Lídia Jorge Tasima, Alexandre Keiji Tashima, Erika Sayuri Nishiduka.

**Supervision:** Karen de Morais-Zani, Anita Mitico Tanaka-Azevedo.

**Validation:** Daniela Miki Hatakeyama, Lídia Jorge Tasima, Kathleen Fernandes Grego, Alexandre Keiji Tashima, Erika Sayuri Nishiduka, Karen de Morais-Zani, Anita Mitico Tanaka-Azevedo.

**Visualization:** Daniela Miki Hatakeyama, Lídia Jorge Tasima, Anita Mitico Tanaka-Azevedo.

**Writing – original draft:** Daniela Miki Hatakeyama, Lídia Jorge Tasima, Anita Mitico Tanaka-Azevedo.

**Writing – review & editing:** Daniela Miki Hatakeyama, Lídia Jorge Tasima, Daniel Rodrigues Stuginski, Anita Mitico Tanaka-Azevedo.

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
