## [Decision Letter · Decision Letter 0]

26 Jan 2021

PONE-D-20-35649

From birth to adulthood: An analysis of Bothrops moojeni snake venom at different life stages

PLOS ONE

Dear Dr. Tanaka-Azevedo,

Thank you for submitting your manuscript to PLOS ONE. After careful consideration, we feel that it has merit but does not fully meet PLOS ONE’s publication criteria as it currently stands. Therefore, we invite you to submit a revised version of the manuscript that addresses the points raised during the review process.

In your revised version please try to address as fully as possible the detailed and constructive comments of the two reviewers.

We look forward to receiving your revised manuscript.

Kind regards,

Israel Silman

Academic Editor

PLOS ONE

Journal Requirements:

2.PLOS ONE now requires that authors provide the original uncropped and unadjusted images underlying all blot or gel results reported in a submission’s figures or Supporting Information files. This policy and the journal’s other requirements for blot/gel reporting and figure preparation are described in detail at https://journals.plos.org/plosone/s/figures#loc-blot-and-gel-reporting-requirements and https://journals.plos.org/plosone/s/figures#loc-preparing-figures-from-image-files. When you submit your revised manuscript, please ensure that your figures adhere fully to these guidelines and provide the original underlying images for all blot or gel data reported in your submission. See the following link for instructions on providing the original image data: https://journals.plos.org/plosone/s/figures#loc-original-images-for-blots-and-gels.

Reviewers' comments:

Reviewer's Responses to Questions

**Comments to the Author**

1. Is the manuscript technically sound, and do the data support the conclusions?

Reviewer #1: Partly

Reviewer #2: Partly

2. Has the statistical analysis been performed appropriately and rigorously? 

Reviewer #1: Yes

Reviewer #2: Yes

3. Have the authors made all data underlying the findings in their manuscript fully available?

Reviewer #1: No

Reviewer #2: Yes

4. Is the manuscript presented in an intelligible fashion and written in standard English?

Reviewer #1: No

Reviewer #2: Yes

5. Review Comments to the Author

Reviewer #1: Major comments

1. Venoms from various ontogenetic stages were collected from snakes belonging to distinct geographical regions. How can the authors be sure that the differences they see are because of ontogenetic shifts, and not because of differences in geographical locations (i.e., intraspecific / interpopulation variation)? Wouldn’t it make much more sense to compare snakes of different ontogenetic stages from the same locality?

2. Male and Female venoms at different stages are represented by unequal sample sizes. Could the effect of unequal pooling influence the overall functional differences?

3. Authors should describe their relative quantification steps in detail. LFQ works best when the spectra are searched against species-specific transcriptomes or when working with model organisms. Performing searches against the Uniprot database, that too by restricting it to a specific clade, is likely to produce erroneous results.

4. The RAW mass spec data should be made available to the readers. Authors should consider depositing this to public repositories, such as the ProteomeXchange

Minor comments

1. Protein concentrations of different samples should be presented in table 1.

2. Line 423 could be Neonate and/or Juvenile snake as haemorrhagic activity was not tested for juveniles in this study.

3. “B. moojeni snake venoms were milked at the Laboratory of Herpetology of Instituto Butantan, according to the laboratory protocol” The authors should clarify what lab protocol they are referring to here.

4. Table 1 is very confusing. It might make it clear to separate the number of individuals from the SVN data.

5. In the materials and methods, the authors are advised to provide catalogue numbers of the chemicals that they have used, which will make it easier for the readers to reproduce these results.

Grammatical errors

The manuscript has many grammatical errors. I have only listed a few below.

1. Line 31: “in specific groups”

2. Suggest rewriting this awkward sentence: “has been studied for many years, as well as individual variation”. Moreover, I fail to understand this point as the individual venom variation is driven by the other reasons they mention in this sentence.

3. “their respective offspring of nine species of Bothrops”

4. “in regards of clinical outcome”

5. “have an impact in snakebite treatment”

6. “until the analyses Table 1 shows the groups, ages, number of”

7. “According to previous works [20,22,63–67], protein families were assigned.”

Reviewer #2: The experimental methodology (assays, LC-MS/MS, SDS-PAGE, etc.) evaluating the venoms of Bothrops moojeni from birth to adulthood is robust, as is the discussion incorporating results from other publications. My major concern is that the pools of venom from different age groups were from snakes of different localities. Therefore, some of the variation trends could be due to geographic variation and not ontogeny. This manuscript might also be better suited in a journal with more specific readership, such as the journal Toxicon.

Minor suggestions for improvement:

Abstract

I feel an additional sentence should be added about the 3 years-old snake venoms, were they similar to juveniles or different from both juveniles and adults? This detail is missing from the abstract currently.

Introduction

The second paragraph is a little short. It would be important to mention what toxins specifically are in this venom and what are the activities of the major toxins (metalloproteinases, serine proteases, etc.).

Line 68: More detail should be added here, such as how was the venom different between the adult females and offspring (what components specifically)?

Material and methods

Any statistics that could also incorporate geographical variation?

Results

Figure 2. It would be nice to see the protein families labeled on the side of the gel. Also, arrows pointing out the differences are that are discussed.

Figure 3. Figure resolution needs to be improved. I cannot see the labels on the chromatograms, including the x- and y-axis, they should be larger and clearer. What does the asterisk signify? This should be noted in the legend.

Line 342: “greatest” concentration

Figure 5: None of these were statistically significant? If so, an asterisk should be added to the corresponding comparison.

Figure 7. Like the other gel, it would be nice to see the protein families labeled on the side of the gel near their size ranges.

Discussion

Line 427: A different word then “besides” should be used. Perhaps begin this sentence listing the authors of the publication?

The discussion is a bit long.

6. PLOS authors have the option to publish the peer review history of their article (what does this mean?). If published, this will include your full peer review and any attached files.

Reviewer #1: No

Reviewer #2: No

---

## [Author Response · Author response to Decision Letter 0]

12 Mar 2021

Reviewer #1: Major comments

1. Venoms from various ontogenetic stages were collected from snakes belonging to distinct geographical regions. How can the authors be sure that the differences they see are because of ontogenetic shifts, and not because of differences in geographical locations (i.e., intraspecific / interpopulation variation)? Wouldn’t it make much more sense to compare snakes of different ontogenetic stages from the same locality?

We would like to thank the reviewer for this inquiry. We understand the concern and we are aware of this issue. Indeed, in an ideal situation, it would be much better to compare the venoms of snakes from the same locality. However, we did not dispose of such material. Furthermore, a previous work by our group (Aguiar et al. Comparative compositional and functional analyses of Bothrops moojeni specimens reveal several individual variations; PloS ONE 14(9): e0222206, 2019) compared individual venoms of 13 snakes from different localities but did not identified a relationship between activities and geographical location. We understand the limitations of working with pooled venoms, but it is what we had available.

2. Male and Female venoms at different stages are represented by unequal sample sizes. Could the effect of unequal pooling influence the overall functional differences?

We thank the reviewer for pointing this out. This difference may have influenced the outcome of some results. However, we collected the venoms from the snakes we had available and we described this limitation in the Discussion (lines 588-600).

3. Authors should describe their relative quantification steps in detail. LFQ works best when the spectra are searched against species-specific transcriptomes or when working with model organisms. Performing searches against the Uniprot database, that too by restricting it to a specific clade, is likely to produce erroneous results.

Thank you for pointing this out. We add more details of the LFQ. We used the three most intense peptide ions to compare the relative abundances of the quantified proteins. We agree with the reviewer that LFQ would work best with species-specific databases. However, the aim of our quantitative approach was only to estimate and compare proteins at the protein family level.

4. The RAW mass spec data should be made available to the readers. Authors should consider depositing this to public repositories, such as the ProteomeXchange

The proteomics data have been deposited to the ProteomeXchange Consortium via the PRIDE partner repository with the dataset identifier PXD024447. 

Submission details:

Project Name: From birth to adulthood: An analysis of Bothrops moojeni snake venom at different life stages

Project accession: PXD024447

Project DOI: Not applicable

Reviewer account details:

Username: reviewer_pxd024447@ebi.ac.uk

Password: KjgIbK2q

Minor comments

1. Protein concentrations of different samples should be presented in table 1.

We thank the reviewer for this suggestion. We added this information to Table 1.

2. Line 423 could be Neonate and/or Juvenile snake as haemorrhagic activity was not tested for juveniles in this study.

Thank you for the suggestion. We replaced “juvenile or an adult” by “neonate, juvenile, or adult” (line 477).

3. “B. moojeni snake venoms were milked at the Laboratory of Herpetology of Instituto Butantan, according to the laboratory protocol” The authors should clarify what lab protocol they are referring to here.

We thank the reviewer for the suggestion. The article describing the Laboratory of Herpetology protocol was added (line 121).

4. Table 1 is very confusing. It might make it clear to separate the number of individuals from the SVN data.

Thank you for the suggestion. We separated number of individual from the average size of the snakes.

5. In the materials and methods, the authors are advised to provide catalogue numbers of the chemicals that they have used, which will make it easier for the readers to reproduce these results.

Thank you for pointing this out. We added this information in Materials and Methods.

Grammatical errors

The manuscript has many grammatical errors. I have only listed a few below.

1. Line 31: “in specific groups”

2. Suggest rewriting this awkward sentence: “has been studied for many years, as well as individual variation”. Moreover, I fail to understand this point as the individual venom variation is driven by the other reasons they mention in this sentence.

3. “their respective offspring of nine species of Bothrops”

4. “in regards of clinical outcome”

5. “have an impact in snakebite treatment”

6. “until the analyses Table 1 shows the groups, ages, number of”

7. “According to previous works [20,22,63–67], protein families were assigned.”

We would like to thank the reviewer for all these suggestions. We revised the manuscript and corrected the mistakes.

Reviewer #2:

The experimental methodology (assays, LC-MS/MS, SDS-PAGE, etc.) evaluating the venoms of Bothrops moojeni from birth to adulthood is robust, as is the discussion incorporating results from other publications. My major concern is that the pools of venom from different age groups were from snakes of different localities. Therefore, some of the variation trends could be due to geographic variation and not ontogeny. This manuscript might also be better suited in a journal with more specific readership, such as the journal Toxicon.

We would like to thank the reviewer for this inquiry. We understand the concern and we are aware of this issue. Indeed, in an ideal situation, it would be much better to compare the venoms of snakes from the same locality. However, we did not dispose of such material. Furthermore, a previous work by our group (Aguiar et al. Comparative compositional and functional analyses of Bothrops moojeni specimens reveal several individual variations; PloS ONE 14(9): e0222206, 2019) compared individually the venoms of 13 snakes from different localities but did not identified a relationship between activities and geographical location. We understand the limitations of working with pooled venoms, but it is what we had available.

Minor suggestions for improvement:

Abstract

I feel an additional sentence should be added about the 3 years-old snake venoms, were they similar to juveniles or different from both juveniles and adults? This detail is missing from the abstract currently.

Thank you for the suggestion. We included a sentence about the 3 years-old snake venoms (lines 32-33).

Introduction

The second paragraph is a little short. It would be important to mention what toxins specifically are in this venom and what are the activities of the major toxins (metalloproteinases, serine proteases, etc.).

We would like to thank the reviewer for the suggestion. We included this information in the Introduction (lines 55-66).

Line 68: More detail should be added here, such as how was the venom different between the adult females and offspring (what components specifically)?

Thank you for the suggestion. We rewrote this part to include this information (lines 90-94).

Material and methods

Any statistics that could also incorporate geographical variation?

We appreciate the suggestion. However, due to sample shortage (especially of younger snakes), the analyses were carried out using pooled venoms, so we do not have specific data of each individual activity or composition. Therefore, we agree with the reviewer that this information could improve our work, but it is not possible to add it.

Results

Figure 2. It would be nice to see the protein families labeled on the side of the gel. Also, arrows pointing out the differences are that are discussed.

We would like to thank the reviewer for this suggestion. We inserted the protein families to the image.

Figure 3. Figure resolution needs to be improved. I cannot see the labels on the chromatograms, including the x- and y-axis, they should be larger and clearer. What does the asterisk signify? This should be noted in the legend.

Thank you for pointing this out. We separated this figure in two parts so it can be easier to see and fit into the page. We also included the information regarding the asterisk.

Line 342: “greatest” concentration

We corrected this mistake.

Figure 5: None of these were statistically significant? If so, an asterisk should be added to the corresponding comparison.

We thank the reviewer for this inquiry. Actually, many groups present statistical differences among each other. To avoid confusion and to keep the image cleaner, we decided to include this information in the Supplementary Table 2.

Figure 7. Like the other gel, it would be nice to see the protein families labeled on the side of the gel near their size ranges.

We would like to thank the reviewer for this suggestion. We inserted the protein families to the image.

Discussion

Line 427: A different word then “besides” should be used. Perhaps begin this sentence listing the authors of the publication?

Thank you for the suggestion. We modified the sentence from “Besides, the appearance of a band weighing ~23 kDa and the change of low molecular weight proteins were shown in B. atrox venom” to “López-Lozano and collaborators observed the appearing of a ~23 kDa band and the change of low molecular weight proteins in B. atrox venom” (lines 482-484).

The discussion is a bit long.

We thank the reviewer for the concern. However, we believe that the discussion contains the necessary information to corroborate the results and the importance of our work.

---

## [Decision Letter · Decision Letter 1]

1 Apr 2021

PONE-D-20-35649R1

From birth to adulthood: An analysis of the Brazilian lancehead (Bothrops moojeni) venom at different life stages

PLOS ONE

Dear Dr. Tanaka-Azevedo,

Thank you for submitting your manuscript to PLOS ONE. After careful consideration, we feel that it has merit but does not fully meet PLOS ONE’s publication criteria as it currently stands. Therefore, we invite you to submit a revised version of the manuscript that addresses the points raised during the review process.

As you will see from their reviews, both reviewers feel that certain important issues still require clarification. In your revised manusript please try to address these points as thoroughly as possible.

We look forward to receiving your revised manuscript.

Kind regards,

Israel Silman

Academic Editor

PLOS ONE

Journal Requirements:

Reviewers' comments:

Reviewer's Responses to Questions

**Comments to the Author**

1. If the authors have adequately addressed your comments raised in a previous round of review and you feel that this manuscript is now acceptable for publication, you may indicate that here to bypass the “Comments to the Author” section, enter your conflict of interest statement in the “Confidential to Editor” section, and submit your "Accept" recommendation.

Reviewer #1: (No Response)

Reviewer #2: (No Response)

2. Is the manuscript technically sound, and do the data support the conclusions?

Reviewer #1: Partly

Reviewer #2: Yes

3. Has the statistical analysis been performed appropriately and rigorously? 

Reviewer #1: N/A

Reviewer #2: Yes

4. Have the authors made all data underlying the findings in their manuscript fully available?

Reviewer #1: Yes

Reviewer #2: Yes

5. Is the manuscript presented in an intelligible fashion and written in standard English?

Reviewer #1: Yes

Reviewer #2: Yes

6. Review Comments to the Author

Reviewer #1: Authors still need to address the following major comments

“but it is what we had available”

I don’ think this is an appropriate response. The authors should acknowledge this shortcoming very clearly in their manuscript.

“However, we collected the venoms from the snakes we had available and we described this limitation in the Discussion (lines 588-600).”

Those lines do not contain any information related to this. Authors need to very clearly explain the shortcomings of this study. Perhaps, in a different section. I am sympathetic to their situation and I do realize its not very easy to do these under ideal conditions. But there is no excuse for not highlighting the obvious shortcomings of the study clearly. This will help the readers judge the results and carve out future directions for research.

Protein quantification

This should also be perhaps highlighted in the drawbacks section. Pleas see reviews by Calvete et al. The fact that LFQ was performed can be ignored as long as this is clearly stated as one of the drawbacks. However, restricting searches against Bothrops is just technically wrong. Please check how many Bothrops sequences are there on Uniprot compared to overall number of sequences. Only the reviewed Bothrops sequences are included on Uniprot, which means that by restricting your searches to these handful of sequences, you are increasing your chances of missing out on a much larger number of hits. Uniprot itself is such a small database so there is no excuse for not searching against the complete database.

Reviewer #2: The revised manuscript has greatly improved. There is still the caveat of using pooled venom from snakes of different localities, but given at this time there is no way to fix this, the authors just need to make it very clear in the writing that this is a major limitation in this work and how the results could be impacted. The resolution of the figures still appears a little blurry, I would just make sure they are sharper for publication.

Some minor spelling/grammar issues are still present, I have pointed out a few of these to be corrected, but the manuscript should be edited again:

Abstract Line 39: “highlights the importance of comprehensive studies to better understand venom variability.”

Line 43: “health”

Line 54: inespecifically?

Line 60: “disrupts blood homeostasis” ? Or maybe a better term to clarify?

Line 65: How come only in some localities? Maybe some details to clarify, such as does venom variability for this species have noted variation in antivenom effectiveness? The way the sentence is currently written (and the first line of the abstract) it seems like that it is only a public health problem in some areas and in others it is harmless.

Line 94: “change” instead of modifies

Fig 3 and Fig 4 titles: change to “male B. moojeni” and “female B. moojeni”

Similar title change for Fig 5 and Fig 7. It is best to use B. moojeni throughout the manuscript to avoid nonspecific common names.

Fig 7. “for”? typo

Lines 584: I would clarify this sentence a bit more. The word “promoter” is associated with gene transcription, and miRNAs are post-transcriptional regulators. miRNAs do not necessarily regulate genes, but the translation of their mRNAs.

7. PLOS authors have the option to publish the peer review history of their article (what does this mean?). If published, this will include your full peer review and any attached files.

Reviewer #1: No

Reviewer #2: No

---

## [Author Response · Author response to Decision Letter 1]

25 May 2021

Reviewer #1: Authors still need to address the following major comments

“but it is what we had available”

I don’ think this is an appropriate response. The authors should acknowledge this shortcoming very clearly in their manuscript.

“However, we collected the venoms from the snakes we had available and we described this limitation in the Discussion (lines 588-600).”

Those lines do not contain any information related to this. Authors need to very clearly explain the shortcomings of this study. Perhaps, in a different section. I am sympathetic to their situation and I do realize its not very easy to do these under ideal conditions. But there is no excuse for not highlighting the obvious shortcomings of the study clearly. This will help the readers judge the results and carve out future directions for research.

We thank the reviewer for this observation. We were not very clear about this shortcoming, so we tried to clarify this issue in the manuscript (lines 575-580).

Protein quantification

This should also be perhaps highlighted in the drawbacks section. Pleas see reviews by Calvete et al. The fact that LFQ was performed can be ignored as long as this is clearly stated as one of the drawbacks. However, restricting searches against Bothrops is just technically wrong. Please check how many Bothrops sequences are there on Uniprot compared to overall number of sequences. Only the reviewed Bothrops sequences are included on Uniprot, which means that by restricting your searches to these handful of sequences, you are increasing your chances of missing out on a much larger number of hits. Uniprot itself is such a small database so there is no excuse for not searching against the complete database.

Thank you for the comment. The reviewer is correct, we have less toxins restricting the database to the reviewed Bothrops sequences. However, as the objective of this database search was to classify toxins only at the family level (and not at the toxin level), we believe we have the main toxins represented in the reviewed database.

Reviewer #2: The revised manuscript has greatly improved. There is still the caveat of using pooled venom from snakes of different localities, but given at this time there is no way to fix this, the authors just need to make it very clear in the writing that this is a major limitation in this work and how the results could be impacted. The resolution of the figures still appears a little blurry, I would just make sure they are sharper for publication.

We thank the reviewer for this observation. We tried to clarify this issue regarding the samples in the manuscript (lines 575-580). As suggested before, we used PACE to optimize the images, but it seems some issues might not have been corrected. We reuploaded the images to PACE and hope this time this matter is resolved.

Some minor spelling/grammar issues are still present, I have pointed out a few of these to be corrected, but the manuscript should be edited again:

Abstract Line 39: “highlights the importance of comprehensive studies to better understand venom variability.”

Line 43: “health”

Line 54: inespecifically?

Line 60: “disrupts blood homeostasis” ? Or maybe a better term to clarify?

Line 65: How come only in some localities? Maybe some details to clarify, such as does venom variability for this species have noted variation in antivenom effectiveness? The way the sentence is currently written (and the first line of the abstract) it seems like that it is only a public health problem in some areas and in others it is harmless.

Line 94: “change” instead of modifies

Fig 3 and Fig 4 titles: change to “male B. moojeni” and “female B. moojeni”

Similar title change for Fig 5 and Fig 7. It is best to use B. moojeni throughout the manuscript to avoid nonspecific common names.

Fig 7. “for”? typo

Lines 584: I would clarify this sentence a bit more. The word “promoter” is associated with gene transcription, and miRNAs are post-transcriptional regulators. miRNAs do not necessarily regulate genes, but the translation of their mRNAs.

We would like to thank the reviewer for all the suggestions. We modified the manuscript accordingly.

---

## [Editor Report · Decision Letter 2]

28 May 2021

From birth to adulthood: An analysis of the Brazilian lancehead (Bothrops moojeni) venom at different life stages

PONE-D-20-35649R2

Dear Dr. Tanaka-Azavedo,

We’re pleased to inform you that your manuscript has been judged scientifically suitable for publication and will be formally accepted for publication once it meets all outstanding technical requirements.

Kind regards,

Israel Silman

Academic Editor

PLOS ONE
---

## [Editor Report · Acceptance letter]

2 Jun 2021

PONE-D-20-35649R2 

From birth to adulthood: An analysis of the Brazilian lancehead (*Bothrops moojeni*) venom at different life stages 

Dear Dr. Tanaka-Azevedo:

I'm pleased to inform you that your manuscript has been deemed suitable for publication in PLOS ONE. Congratulations! Your manuscript is now with our production department. 

Kind regards, 

on behalf of

Prof. Israel Silman 

Academic Editor

PLOS ONE